# Current Trajectories and New Challenges for Visual Comfort Assessment in Building Design and Operation: A Critical Review

Juan Diego Blanco Cadena [1,*], Tiziana Poli [1], Mitja Košir [2], Gabriele Lobaccaro [3], Andrea Giovanni Mainini [1] and Alberto Speroni [1]

1   ABC Department, Politecnico di Milano, 20133 Milan, Italy; tiziana.poli@polimi.it (T.P.); andreagiovanni.mainini@polimi.it (A.G.M.); alberto.speroni@polimi.it (A.S.)
2   Faculty of Civil and Geodetic Engineering, University of Ljubljana, 1000 Ljubljana, Slovenia; mitja.kosir@fgg.uni-lj.si
3   Department of Civil and Environmental Engineering, Faculty of Engineering, Norwegian University of Science and Technology, Høgskoleringen 7a, 7491 Trondheim, Norway; gabriele.lobaccaro@ntnu.no
*   Correspondence: juandiego.blanco@polimi.it

**Abstract:** Visual comfort can affect building occupants' behaviour, productivity and health. It is highly dependent on the occupant and how they perform a task indoors. In that regard, an occupant centred approach is more suitable for evaluating the lighting perception of the indoor environment. Nevertheless, the process of rating and estimating the visual comfort makes a limited distinction between physiological differences (e.g., ageing eye, light sensitivity), field of view, and personal preferences, which have been proven to influence the occupants' lighting needs to complete their tasks. Such features were not considered while establishing the visually comfortable conditions; perhaps due to the challenge of coupling the assumptions made during building design to the performance indicators monitored during building operation. This work focuses on reviewing literature findings on how the common design approach deviates from real building performance, particularly failing to prevent visual disturbances that can trigger the inefficient operation of building systems. Additionally, it is highlighted that redesigned visual comfort assessment methods and metrics are required to bridge the gap between the lighting environment ratings computed and surveyed. One possibility is to consider such physiological features that induce lighting experiences . Finally, it was deduced that it is important to target the occupants' eye response to calibrate limit thresholds, propose occupant profiling, and that it is convenient to continuously monitor the occupants' perception of indoor lighting conditions.

**Keywords:** visual comfort; lighting; daylight; user centred design; modeling; monitoring

## 1. Introduction

The concept of a building was initially used to identify a human shelter (i.e., enclosure) that protects its occupants from weather, and is built for security as a livable and private space. The understanding of the concept was later expanded to also include the notion of a comfortable living and working space. Nevertheless, the notion of comfort has been interpreted in numerous ways. In the beginning, the building was understood as somewhere that had sufficient weather protection and security from any outdoor threats, and later evolved into a healthy space in an unperturbed state; thus, making the indoor environment the space where humanity spends most of its time (>80%) [1,2]. The indoor environment can be categorized as an anthropogenic artificial habitat, in which, at least to some degree, its conditions can be continuously monitored by sensors and controlled by integrated building systems. These conditions were envisioned to promote comfort, health, and well-being for most building occupants. Consequently, ranges of physical parameters that describe the indoor environment (i.e., air temperature, air humidity, illuminance) were

established in such a way that, if respected, would guarantee satisfactory indoor conditions, promote the reduction of undesired human body responses (e.g., onset of sick building syndrome) and enhance occupants' quality of life [3,4].

To yield such a controlled indoor environment, active building systems are unavoidable (e.g., air handling units, lighting appliances), which constitute a significant energy utilization rendering the building sector responsible for a significant share of the worldwide energy consumption and $CO_2$ emissions to the atmosphere [5]. Therefore, to contrast with current trends, it is paramount to understand which are the drives for such energy consumption and $CO_2$ emission growth, and their links with the building users' comfort, health, and well-being.

Currently enforced building design policies [6,7] tackle climate change by encouraging the realization of highly resilient, sustainable and energy efficient buildings with low carbon footprint and reduced energy intensity while, sustaining occupant satisfaction with the indoor environment. In this regard, new and restored building quality has been enhanced. This is reflected in a substantial increase of certified buildings, either by national standards, organizations (e.g., Passive House, US Green Building Council) or public bodies/institutions (e.g., BREEAM buildings). Such certification systems give great importance to green strategies related to embodied energy and $CO_2$ emissions of materials, installation of efficient appliances, and promotion of green behaviour of buildings' users. Nevertheless, they downplay the relevance of considerations taken regarding occupants' comfort, health and well-being [8]; even when a building's overall climate burden is more intense during its operation phase (i.e., delivering set indoor conditions). Besides, the actual building energy performance has deviated from the desired designed performance (see the case of the European Union (EU) deconstructed in Figure 1 and further explained in Section 3.1), which has been partly attributed to the users' interactions about the planned building functioning [9].

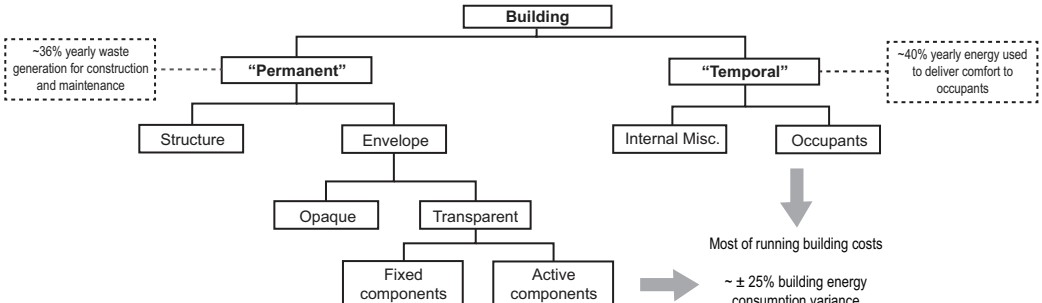

**Figure 1.** Building components deconstruction outline, and building-occupants interaction potential outcome on climate burden. Based on data collected for the European context.

Hence, if the efficiency of energy use is interpreted as managing it appropriately to serve the occupant, it is necessary to better understand the preferences and needs of the individual as well as of the group of occupants. This is still an unresolved matter, even when comfort ranges were established with a high level of acceptance (see Section 4.1). However, having a better suited definition of true comfort conditions based on a more representative occupancy type would allow superior accuracy in monitoring building operation and design [9–11].

In this framework, the presented work has concentrated on, firstly, collecting literature insights to understand why building designers and facility managers are not succeeding in narrowing the gap between modelled and actual building performance (i.e., performance gap) and secondly, highlighting the importance of acknowledging the building systems capabilities, the occupants' real comfort needs and environmental conditions. While finally, proposing potential solutions to improve the accuracy of comfort assessment which can boost building operation performance by reducing unpredictable occupant–building interactions.

## 2. Literature Collection Criteria

To capture the aforementioned needed understanding, literature was collected in different bundles. Firstly, literature was gathered referring to actual building operation performance, and how closely is it aligned with the expected performance estimated during the building design phase. Consequently, information was screened to identify the main reasons why such performance gap exists, concentrating in the factors that can be further assessed by scientist and designers. The magnitude of the influence of the effect of one of the identified factors was grasped (i.e., visual comfort perception variance) to support and recall the importance of its evaluation (Section 3).

Secondly, literature has been collected and analyzed on different methods applied for optimizing lighting-related building design and operation. In this context, literature was collected with three purposes: (i) gather, analyse and identify the most frequently used metrics, indexes or KPIs, to evaluate and/or describe the potential perception that a building occupant could have of the provided indoor luminous environment (Section 4); (ii) compare the metrics based on the guidelines which have embraced them, their strengths, weaknesses and challenges (Section 4.1); and (iii) cluster the different strategies employed using such metrics for optimizing building shape, building systems and building operation (Section 4.2).

Finally, based on the challenges identified on the most commonly utilized KPIs and followed lighting design approaches, literature on distinctive visual comfort assessment methods are summarized, presented and described as potential solutions to the identified gaps (Sections 5 and 6). Solutions are presented for different phases of the study of building performance (i.e., design and operation), highlighting their contribution and the current barriers that could be addressed in the future.

## 3. Motivation for Occupant-Centered Assessment

Sufficient evidence has been documented to assume that the current visual comfort assessment approach considers only marginal, or tends to underestimate, the effect of influencing factors resulting in unreliable comfort provision design/operation and thus, higher variance between the expected and the actual user experience of the indoor luminous environment [12–19].

### 3.1. Effect of Occupant Interactions on Building Performance

Generally, in the building design phase for visual comfort assessment, occupancy is considered on the level of a standard occupant (following an occupancy schedule and space type), with an idealization of their average conditions (i.e., healthy, average age and height). Sometimes also additional parameters are set, like, a trigger for shading devices (i.e., shading area ratio) depending usually on the designers' defined optimum indoor temperature and more really on the set lighting conditions (i.e., monitoring illuminance level and radiation intensity) [20].

However, building occupants should be contemplated more holistically and as active participants. If they are dissatisfied with the delivered indoor environment, they will interact with the building and modify its operation, resulting in altered performance (Table 1). In addition, occupants can tweak the building performance for the non-performed actions within buildings lacking automation (e.g., forgetting to turn off lighting appliances or deactivating shutters) when the surrounding conditions would have provided a sufficiently comfortable environment without an intervention of a building system (e.g., shading device, lighting appliances) or when occupants leave the previously occupied space [21]. In this context, Kamaruzzaman et al. [15] surveyed occupants in different refurbished buildings in Malaysia, finding that daylighting, electric lighting, and glare were among the most dissatisfying aspects of the building's operation and most of the time the blinds were down to avoid indirect disturbing glare from surrounding surfaces (i.e., computer screens) incurring in higher lighting energy use. Likewise, Masoso and Gobler [9] presented an example in which this misinteraction between occupants and building systems, led

to an extreme of more than 50% of a building's energy use. This excess mainly happened during non-occupied hours, identifying that lighting appliances are ranked second among the electricity waste in the case of a university building. On the contrary, Reinhart et al. [22] screened day-lit offices in Germany and concluded that building occupants would have a more positive disposition for turning off lighting appliances and blind modification towards a brighter environment with no glare risk, rather than a darker indoor space, which could result in higher cooling loads.

**Table 1.** Summary of the effects (positive [+] or negative [−]) of occupant–building interaction on building energy performance.

| Interaction Type | Effect | Description | References |
| --- | --- | --- | --- |
| Blind operation | + | Occupants tend to open blinds to favor brighter indoor environments without glare risk. Which often leads to lower energy use, when solar radiation is not too intense. A manual control over lights and blinds could produce annual reductions in lighting loads up to 80% compared to a constant lighting activation. | [22,23] |
| | + | Occupants tend to close blinds in instances of incident outdoor illuminance above 50 klux or to avoid direct sunlight above 50 W/m$^2$. Protecting from both excessive light and radiation influx. | [22] |
| | +/− | Occupants tend to open blinds to favor brighter indoor environment without glare risk. Having a larger glare tolerance, or avoiding glare by different means, could lead to higher cooling (or lower heating) loads compared to estimated performance. | [22] |
| | − | Occupants tend not to open the blinds to increase daylight influx when sufficient outdoor illuminance is present. Which often leads to larger energy use to compensate for the lack of adequate illuminance levels. | [15,21,24] |
| | − | Occupants tend to close the blinds upon arrival regardless of the outdoor illuminance intensity, even when proper or insufficient for indoor lighting. This often leads to larger energy use of lighting appliances. | [21] |
| | − | Glare control measures can negatively affect the illuminance level and illuminance uniformity in an office. | [25] |
| Light appliances use | + | Automatic daylight linked dimming of lights is acceptable to occupants. Although, manual dimming results in higher occupant satisfaction levels. Moreover, these systems motivate occupants to use more daylight. | [24,26] |
| | + | Active automated lighting control (ON/OFF light switching with ideal photocell-based dimming and occupancy-sensing OFF switching) could lead to an annual reduction in lighting loads up to 90% if compared to a constant lighting activation scenario. | [23] |
| | + | For most users the decision to turn lights on when arriving depends on the daylight level in the room at that moment since, on average, occupants switch electric lights on more frequently in the case of low daylight induced illuminances. | [27,28] |
| | + | Manual control over lights and blinds could lead to an annual reduction in lighting loads up to 80% if compared to a constant lighting activation scenario. | [23] |
| | + | Passive reminders (i.e., stickers installed by the light switches) substantially increase the occupants' turning off activity. Avoiding energy waste | [29] |
| | − | Forgetting lighting appliances on, when not needed can lead to an excess up to 50% of energy use. Indeed, up to 90% of manual controls occur just after the occupant enters the room or just before they left. | [9,27] |

## 3.2. Occupant-Building Interaction Asymmetry

As indicated in the previous section, and depending on the occupancy type held in the building (see Section 5), the occupant–building interaction can be towards opposite preferences, and in some cases, decisive on building operation (Table 1). In fact, unde-

sired human–building interaction, or idleness, can lead to 25% variation in yearly energy consumption compared to traditional building energy modelling (BEM) (Figure 1) [30].

Such variance can be attributed to the physical conditions which affect occupant perception, hence deviating from the standard visually comfortable ranges. People could suffer from eye sight issues, or can be significantly affected by the ageing of the eye, demanding more specific lighting provision requirements; few may be more sensitive to high illuminance levels due to their eye colour or to particular light colour given their eye physiology. Whereas, some others with the use of contact lenses or glasses, might be more or less tolerant to low or high illuminance or luminance levels depending on the case. Finally, other issues can significantly influence the comfort ranges or limits, on which building occupants would provoke (or would have liked) interference with the programmed building operation [11,31–36].

Unfortunately, it is not only about the physical conditions, but also mentality, awareness, education, or habits that can modify the indoor environment perception and rating [11,14,36–38]. Currently, energy modelling and regulations do not include some of these factors for relevant updates that would guide designers and facility managers towards better buildings for clients and users.

Although researchers have worked on predicting the occupancy effect (e.g., the number of occupants present, interaction trends with building systems) [30,39–49], the accuracy is still unsatisfactory. Some relevant factors are still undermined, and part of them are closely related to the type of occupants in the building, affecting comfort perception and more frequently triggering their interaction with the building. Detailed analysis of the occupancy type in buildings is rarely conducted, even when some building design guidelines include an optional methodology [50]. This happens mainly due to the complexity and the actual research extent of such a process.

Specifically for indoor visual comfort, Pierson et al. [36] presented a collection of environmental and occupancy related factors that are currently under scrutiny to improve visual comfort assessment and the extent to which they have been studied and proved to influence the perception of the indoor environment. Those aspects which are closely linked to different occupant preferences and physiological needs are described further in Section 5.

Such contrasting results, together with the potential positive increment in occupants' mood, satisfaction and productivity alongside lower building energy consumption [51], motivated this work to concentrate on how the scientific community is dealing with this discrepancy, generated by the significant deviation of occupants' preferences in the indoor lighting environment. Specifically, the focus is on how to better understand the drivers of users' interactions with the building aiming to reduce performance deviations.

### 3.3. Magnitude of Occupant Variability

The type of occupancy to be considered in building design and their comfort settings for building operation are set for healthy occupants of a specific age range and race or origin. These might not be in accordance with the actual or designated occupancy of the studied building. For instance, ISO2004 [52] has been defined as standard occupants: 35 year old, male and female, which vary slightly in height (1.70 and 1.60 m), weight (70 and 60 Kg), body surface (1.80 and 1.60 $m^2$) and basal metabolic rate (44 and 41 $W/m^2$). Such assumptions might be misleading, as the age and height (which have been reported sensitive for visual perception variance [32,36]) could vary significantly depending on the building function type and the context within which the project is placed.

In addition, while in the instance of thermal comfort, a body-temperature regulation model has been included to predict thermal comfort perception (e.g., Predicted Mean Vote), on the contrary, an analogous model for visual comfort is still missing. A procedure is needed to include the physiological features of occupants to assess visual comfort, to predict visual perception or luminous environment rating.

Researchers have exposed noticeable differences in occupants' eye adaptation capacity, physiological diversity, personalities, education, income and even attitude. Sustainable

awareness among the world population is increasing, average population age is growing, as well as the number of people with eye-sight impairment (World Health Organization, https://www.who.int/news-room/fact-sheets/detail/blindness-and-visual-impairment (accessed on 5 November 2021)). Simultaneously, the income per capita and education level are diverse among world regions (Figure 2). The exposed demographic characteristics will affect occupants perception and rate of the indoor luminous environment, claiming an update to the current established visual comfort assessment methodologies for buildings.

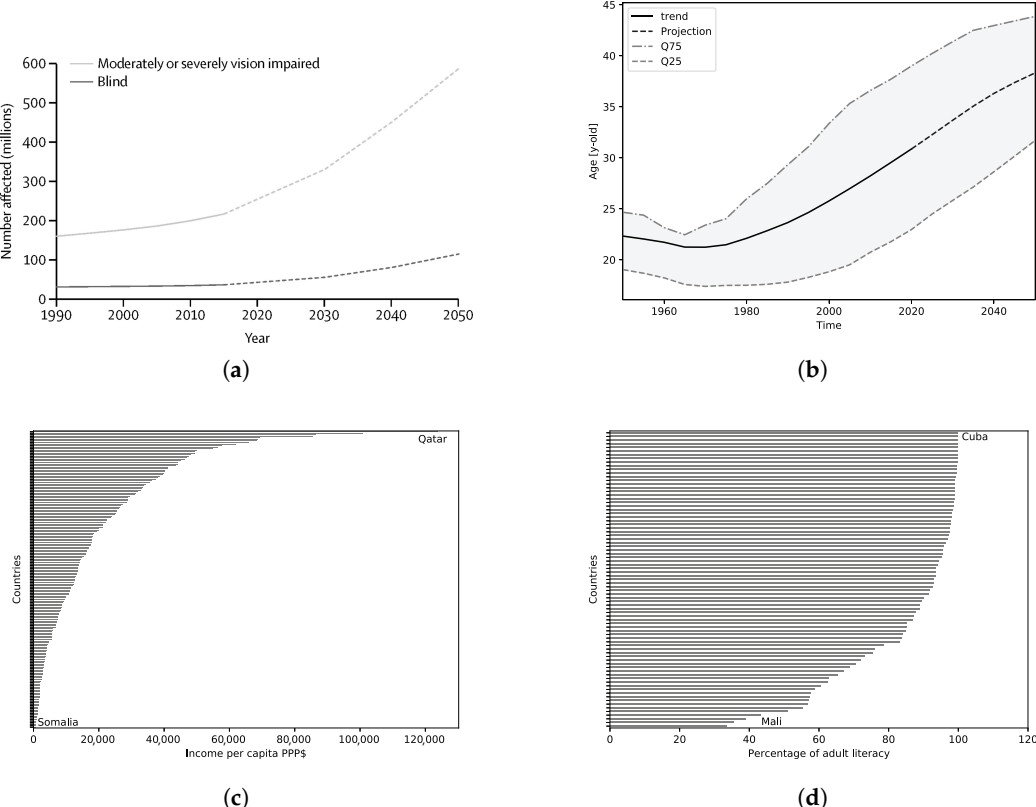

**Figure 2.** World's population differences probably affecting indoor comfort perception. In particular, that could skew the results obtained through qualitative questionnaires. (**a**) Eye-sight affections (extracted from [53]). (**b**) Population average world's age trend; (**c**) Projected 2020's per capita Purchasing Power Parity (PPP) differences; and, (**d**) 2011's Percentage of adult literacy breach based on free material from GAPMINDER.ORG (accessed on 23 December 2019) , CC-BY LICENSE.

## 4. Evaluating the Indoor Luminous Environment and Visual Comfort Perception

The indoor luminous environment is the result of the interaction between the building surrounding context and the integrated strategies planned during the design phase. Building passive and active performance strategies can be summarized in optimizing building shape and orientation, glazing and shading properties, integrating and enhancing dynamic (i.e., climate responsive) shading devices and artificial lighting appliances. Upgrading the performance and operation of passive and active strategies, as well as boosting their interoperability, would significantly render a building more efficient in energy use and comfort provision. Accordingly, it could be derived from the identified occupants' preferred indoor conditions to formulate an efficient control system algorithm. However, the proper performance criteria needs to be selected; that is, a comfort-based performance indicator. And, given the variability presented in Section 3.2, visual comfort occupant-centred approaches would be preferable.

Indicators can be either a unique indicator that communicates the state of a particular parameter, one that incorporates multiple parameters to rate an overall condition, or one that comprises multiple contemporary effects on the occupant. However, when

considering only the environmental parameters, and based on the robust review made by Pierson et al. [36], the selected performance criteria for an accurate occupant-centred visual comfort assessment should undoubtedly account for:

- luminance of the glare source;
- illuminance on the eye-plane;
- adaptation level;
- contrast effect;
- size of the glare source;
- position of the glare source.

While, it should likely consider:

- saturation effect;
- light resulting spectrum;
- light colour temperature.

The impact of the quality of the view through the window has not yet been deciphered by a comprehensive research approach. Nevertheless, according to Ko et al. [54], the view quality can be defined as a combination of view content (i.e., what can be seen), view access (i.e., the amount of view seen by the occupant) and view clarity (i.e., how clearly the content can be seen due to the properties of the window).

### 4.1. Visual Comfort Evaluation Metrics

To estimate the comfort degree of occupants, or to understand if they are comfortable with its surrounding lighting environment, many indicators have been proposed and numerous have been established within the design regulations and guidelines.

Visual comfort is mainly evaluated based on the intensity of light falling on a surface (i.e., illuminance, $E$), either vertical ($E_v$) or horizontal ($E_h$), measured in lux. Thus, the visual comfort metrics are generally based on the light intensity, distribution, source and directionality. The quantity and quality of light should be enough to allow building users to perform the assigned area's correspondent activities safely and without induced visual fatigue. Artificial lighting should only provide the missing light to reach the minimum requirement (specific for each task) only when natural light is not intense enough, aiming to avoid the development of eye sight diseases.

Depending on the regulations governing the design and the building's intended use, the lighting requirements can be evaluated by different indicators and acceptable thresholds or ranges of such. Nevertheless, regardless their time domain, they can be summarized in two main groups: (i) quantity and uniformity of light (Section 4.1.1) and (ii) direct light discomfort (Section 4.1.2).

#### 4.1.1. Quantity and Uniformity of Light

Considering mainly the luminance ($L$) of the light sources and the ambient distribution that condition the actual quantity and uniformity of light falling on a surface, the following metrics have been widely used to express the lighting quality of an indoor space:

- **Daylight Factor (*DF*):** compares the amount of light inside with the exterior during an overcast sky condition [55], the average *DF* across the analyzed surfaces of the building is typically requested to be greater or equal to 2% [56]. Alternatively, the appropriate *DF* value can be defined concerning the specified target illuminance and geographic location, following the method proposed in EN 17037 [57].
- **Daylight Autonomy (*DA*):** represents the percentage of occupied time in which a point within the analysis grid is over a defined minimum illuminance (discrete values only, e.g., complies or not). Additionally, the percentage can also include partial values when the conditions are below the established minimum threshold (ratio between obtained and minimum threshold), and are expressed as **continuous *DA* (*cDA*)** [58].

- **Spatial *DA* (*sDA*):** gives a notion on how much area is over the minimum illuminance for more than a certain portion of the occupied time, typically 300 lux is used for basic reading and writing activities, and these values shall be met for at least 50% of the analysis grid area [59].
- **Useful Daylight Illuminance (*UDI*):** calculates the percentage of the analysed time in which the illuminance in a point falls in a certain established range (normally between 300 and 3000 lux) [60]. Subcategories are also used to communicate the time in which the illuminance values falls below (underlit) or above the suggested ranges (over lit). Specifying the over lit areas can also suggest direct light discomfort risk.
- **Frequency of Visual Comfort (*FVC*):** percentage of time within the analysis period during which appropriate values of average illuminance are accomplished [61]. Similar to UDI, proposes 3 ranges: Comfort, Under and Over.

$$FVC = \frac{\sum_i wf_i t_i}{\sum_i t_i} \tag{1}$$

$$wf_i = \begin{cases} 1 & E_{Under} \leq E_{Daylight} \leq E_{Over} \\ 0 & E_{Daylight} < E_{Under} \, or \, E_{Daylight} > E_{Over}, \end{cases}$$

where:

$E_{Under/Over}$: minimum/maximum illuminance threshold
$E_{Daylight}$: computed/measured illuminance value
$t$: analysis period

- **Intensity of Visual Discomfort (*IVD*):** time integral of the difference between the spatial average of the current daylight illuminance and the upper limit of visual comfort or the lower limit of visual comfort [61].

$$IVD = \int_p \Delta E(t) dt \tag{2}$$

$IVD_{Over}$

$$\Delta E(t) = \begin{cases} E(t) - E_{Over} & E(t) \geq E_{Over} \\ 0 & E(t) < E_{Over} \end{cases}$$

$IVD_{Under}$

$$\Delta E(t) = \begin{cases} 0 & E(t) > E_{Under} \\ E_{Under} - E(t) & E(t) \leq E_{Under}, \end{cases}$$

where:

$E_{Under/Over}$: minimum/maximum illuminance threshold
$E_{Daylight}$: obtained illuminance value
$t$: analysis period

- **Daylight Uniformity (*U_o*):** ratio in a given moment, or time frame, between the minimum value of illuminance on the analysis area ($E_{min}$) and a reference value (e.g., maximum or average) [62].

### 4.1.2. Direct Light Discomfort

Direct light discomfort is commonly referred to as glare. It is meant to quantify the disturbance that a building occupant might perceive based on how the occupants' location and view/sight are exposed to excessive light. It mainly considers the $L$ intensity, concentration, location and the effect of its generated contrast. It can be assessed either directly or indirectly:

- Directly with **Daylight Glare Probability (*DGP*):** which represents the percentage of people that would be disturbed by the level of vertical illuminance ($E_v$) at eye level and the contrast of luminance sources within the occupants' field of view (defined

by [63]). However, it is not valid for $DGP$ values below 0.2 or $E_v \leq 380$ lux; thus, to extend its usability range, an s-curve corrective factor was introduced to compute the glare probability when $E_v$ varies between 0 and 300 lux ($DGP_{low}$) [63].
$DGP$ can also be computed using validated simplified procedures:

–  Using direct correlation with $E_v$ on a vertical plane corresponding to the occupant's eyes height, lo location and orientation, neglecting the contribution of local quantities as presented by [64,65]. In Equation (3), the probability was re-calibrated with the vertical illuminance values only estimated from rendered images [64], while Equation (4) validated in a virtual environment a simplified $DGP$ definition computing illuminance values only using ray-tracing methods (contribution of contrast is neglected).

$$DGP_{Wienold} = 6.22 \times 10^{-5} \times E_v + 0.184 \tag{3}$$

$$DGP_{Hviid} = 5.87 \times 10^{-5} \times E_v + 0.16. \tag{4}$$

–  Using correlation with the computed $L$ from a radiance [66] simulated fish-eye image captured for the desired location, eye-height and gaze orientation using tools such as *evalglare* to compute it [63].

- Glare can also be directly computed using the **CIE Glare Index (*CGI*):** born as a correction of the British Glare Index (*BGI*), presented by Einhorn [67]; and then upgraded into the **Unified Glare Rating (*UGR*)**. It is a short-term, local and one-tailed glare index based on the split contribution of the direct and diffuse $E_h$.
- Finally, glare can also be addressed indirectly by identifying the areas of a building which are subjected to $E_h$ levels over certain threshold, that is either:

    –  Over an upper $E$ comfort threshold (>2000 ÷ 3000 lux is typically used), which can be described using the overlit portion of $UDI$;
    –  Over a maximum $E$ comfort threshold, but sustained for a determined number of hours, as the defined in the case of **Annual Sun Exposure (*ASE*)**, indicates the possibility of glare occurrence (>1000 lux over 250 h per year, is often used) [59].

Tables 2 and 3 represent a comprehensive summary of the guidelines that have embraced the aforementioned metrics, together with their advantages and disadvantages. These metrics are generally utilized and monitored (if possible) on visual comfort assessment methods under a high degree of building design development, or level of detail (LOD).

Nevertheless, all the analysed metrics and suggested values, already established and included in different regulations and design guidelines, do not consider the adaptation level (or capacity) of the eye, nor the variability of occupant's sensibility to light. Thus, they are inaccurate and occupant-centred. Moreover, there are no clear suggestions or procedures developed on how to include them in the building operation phase when monitoring visual comfort (which in some cases is unpractical). Consequentially, a need to further include the occupants' true perception and response to visual stimuli within the loop of building performance design and operation strategies is clearly identified.

**Table 2.** Critical analysis summary of collected visual comfort condition metrics based on daylight intensity and distribution, highlighting their benefits and drawbacks.

| Metric | Guidelines | Advantages | Disadvantages |
|---|---|---|---|
| $E_h$ | LEED, BREEAM, WELL, EN12464-1, EN17037 | Easy to monitor, measure and model; spatial lighting conditions provided. | Point in time dependent, analysis grid selection ambiguous, no human adaptation, light intensity at eye uncertain. |
| $E_v$ | WELL | Easy to measure and model, light intensity at eye known. | Point in time dependent, analysis grid location ambiguous, no human adaptation, complex to measure with occupancy, no contrast effect considered. |
| $DF$ | BREEAM, BS8206-2, DGNB, DIN5034-1, EN17037 | Fast and easy assessment; spatial conditions provided. | Unique low light intensity condition evaluation, no directionality assessed. |
| $DA$ | n/a | Annual and spatial analysis, easy to model. | Dichotomous variable (1 or 0), No high light intensity risk considered, light intensity at eye uncertain, biased by the limits set. |
| $cDA$ | n/a | Annual and spatial analysis, easy to model. Ordinal variable (from 0 to 1). | No high light intensity risk considered, light intensity at eye uncertain, biased by the limits set. |
| $sDA$ | LEED | Annual and spatial unitary index, easy to model. | No high light intensity risk considered, light at eye uncertain, no information on problematic areas, biased by the limits set. |
| $UDI$ | n/a | Annual, grid-based and spatial index, easy to model, hint on under-lit, day-lit and over-lit areas. | Light at eye uncertain, no directionality assessed, biased by the limits set. |
| $FVC$ | n/a | Annual and grid-based unitary index based on $E_h$, easy to model, information on the amount of time at discomfort and comfort. | Light at eye unknown, spatial distribution not known, human adaptation or variability not assessed, biased by the limits selected. |
| $IVD$ | n/a | Annual and grid-based unitary index based on $E_h$, easy to model, information on the intensity of discomfort. | Light at eye unknown, spatial distribution not known, human adaptation or variability not assessed, biased by the limits selected. |
| $U_o$ | AS1680, DIN5035, NSVV, CIBSE, EN12464-1, CIE29.2, BS8206-2, WELL | Information on lighting distribution contrast. | Biased by extreme values, no other information on light, has to be coupled with another metric. |

**Table 3.** Critical analysis summary of collected visual discomfort condition metrics based on daylight intensity and directionality, highlighting their benefits and drawbacks.

| Metric | Guidelines | Advantages | Disadvantages |
|---|---|---|---|
| $DGP$ | EN17037 | Robust and reliable, considers both light intensity and contrast, accounts for directionality, gives rating and sensation information, light at eye known. | Complex and lengthy to model, not reliable under low illuminance conditions and less reliable with large contrast, unique moment in time and position, no human adaptation considered. |
| $DGP_s$ | EN17037 | Still robust and reliable, accounts for directionality, gives rating and sensation information, easy to model and compute, can be computed annually. | No contrast considered, less reliable under low illuminance conditions, analysis grid selection ambiguous, no human adaptation considered. |
| $UGR$ | CIE177, EN12464-1 | Accounts for light intensity, contrast and directionality, gives rating and sensation information, direct and diffuse light at eye known. | Unique moment in time and position, no human adaptation considered, not reliable when the sun is within the field of view. |
| $ASE$ | LEED | Annual and spatial unitary index, easy to model, hint on high intensity problematic areas. | Light at eye uncertain, no directionality assessed, biased by the limits set. |

### 4.2. Current Lighting Design Assessment Strategies

In fact, most simplified design procedures are applied to estimate visual comfort by monitoring only $E_h$ or $DF$. More granular analyses have been proposed showing better results in pursuit of enhanced building occupant conditions and efficient energy utilization. These depend on the building design LOD and the calculation tools' capabilities [68], or the buildings' smart readiness [69], especially as the higher the LOD is, the easier it is to individuate where occupants would be spending most of their time.

#### 4.2.1. Assessment Methods under High Building Design LOD

Assuming that the building systems are not yet defined, but the LOD of the building design phase is already high (LOD approximately between 300–400), the indoor visual

comfort assessment is more likely to result in more accurate results (i.e., realistic) while using the metrics previously presented. Therefore, under the known layout of the points of interest (e.g., workstations in office buildings, study desks in school buildings), or entire areas (e.g., living area in residential buildings):

- $E_h$ is monitored at the task area of each occupant or the most probable occupied area.
- $U_o$ the variability of the light intensity at the task area is compared for each occupant or the most probable occupied area.
- $E_v$ is monitored at the eye level of each occupant position, resembling illuminance at the pupil ($E_p$), for one prevalent line of view or more than one direction.
- $DGP$ is monitored at each occupant position or glare-risk locations, for the most critical moment of the year (based on designer's criteria) or on the location where and point-in-time when higher illuminance intensity was identified (either through $E_h$ or $E_v$).
- $DGP_S$ monitored at each occupant position, computed from estimated $E_v$ values.

### 4.2.2. Assessment Methods under Low Building Design LOD

When building systems and occupant space distribution is unknown or unreliable, detailed analyses are hindered and more robust approaches are generally applied. Some metrics become less effective (e.g., $E_h$ and $U_o$) as some regions of the potentially occupied space could be dedicated to transitional areas (e.g., corridors) and not as task areas. Among literature, the following were identified to have significant potential to better address visual comfort from an occupant-centred perspective when low LOD models are available:

- A modification of the process to rapidly estimate yearly point-in-time glare with $DGP$ from rendered HDR images was presented by Liu et al. [70]. It entails yearly predicted HDR images using deep neural networks from rendering only 5% of the analysis period for every line of view. This methodology showed decent accuracy compared to the results produced by Radiance - *rpict* function, enabling faster yearly calculations of image-based metrics (e.g., $DGP$).
- To spatially cover the visual experience and map the daylight glare class in the room on an annual basis, Giovannini et al. [71] proposed setting lower, intermediate and higher threshold limits of $E_v$ that $DGP$ would rate glare as imperceptible, perceptible and disturbing. These threshold limits are meant to be specific, as they are extracted after identifying the worst condition in the analysed room. The latter is mainly imposed as an occupant located close to the window and with a line of view perpendicular to the window surface plane (few initial point-in-time and image-based simulations are expected).
- To reduce computer simulation time while performing point-in-time and spatial analysis, optimized simulation workflows have been proposed and validated. These optimizations comprise the use of cloud computing services and the integration of graphics processing units (GPUs) [72] . For instance, Ladybug tools [73] has created a new simulation platform service based on cloud computing (i.e., Pollination [74]) able to speed up the simulation process. In particular, when a large number of design options should be tested by running them all in parallel. Jones and Reinhart [75] have developed a suite of GPU-enabled tools that implement ray-tracing functionality that speed-up both Radiance and DAYSIM process by one order of magnitude.
- To avoid extensive computer simulations for monitoring glare and lighting sufficiency, more efficient simulation workflows have been proposed. These have managed to supersede the use of rendered images. For instance, Jones [72] presented a method to fast-compute glare (133,000 times faster) based on the calculation of view factors to a discretized sky dome for estimating direct lighting. Alternatively, cubic illuminance can be computed and re-elaborated [76]. Using $E_h$ and $E_v$ to determine lighting sufficiency and computing $DGP_S$, does not require image-based-point-in-time simulations. A cube-grid-based illuminance simulations with the centre on the hypothetical location of the occupant. It will be directional if ray-tracing engines are used (e.g.,

Radiance). It can be run on an annual basis and, from this, by specifying a direction of view, eye illuminance (i.e., $E_v$ or $E_p$) and work-plane illuminance can be directly computed ($E_h$). Thus, a room can be spatially populated with cubes for mapping all conditions within a room, and posterior data post-processing by exploring multiple view directions can be carried out. This procedure can be carried out more quickly by using a practical approach presented by Cuttle [77]. Otherwise, a much simpler procedure was proposed by Raynham [78] by employing the mean indirect cubic illuminance (MICI).

However, none of the exposed modelling methodologies found were related to the variance of occupants' visual experience, resulting in different comfort perceptions, sensations and ratings. None of them has included the physiological or anatomical variations on subjects that would largely alter the subjects' ratings of a given luminous environment. Moreover, these methodologies become unpractical or infeasible to be surveyed within an existing building. This is because no actual occupants' field of view can be obtained without intrusive methods, and no unbiased visual comfort rating can be obtained without considering further occupants' inherent features.

## 5. Challenging Methods to Include Occupants Visual Comfort Perception Variance

Various comfort indicators have been proposed and established. Nevertheless, the main problem remains how to address the continuous and large variance of occupants' comfort preferences, without largely deviating by applying robust approximations.

Yamin Garreton et al. [79] highlights that "*DGP* models have some limitations for predicting glare in sunny climates with high luminance contrasts", arguing that occupants could have higher glare tolerance, increasing the visual comfort variance. In fact, Frontczak and Wargocki [14] found from a survey analysis that visual comfort perception could be modified by age and type of job or activity executed. According to Bitsios et al. [34] findings, older people's pupil behaviour tends to dilate slower than in the case of younger people but constricts faster. In addition, Goncharov and Dainty [80] presented the relevant effect of ageing on the anatomical structure of the human eye, in particular for the cornea, the lens' thickness and lens' anterior radius; thus, modifying the eye's light regulation capacity.

Finally, Pierson et al. [36] highlighted the degree of certainty of the effect of some of these factors on the occupants' visual perception:

- gender and optical correction influence to be most certainly null;
- age, self-glare assessment, iris pigmentation impact to be yet inconclusive;
- culture, somewhat likely;
- macular pigment optical density (age-dependent), cortical hyperexcitability and contrast sensitivity influence have been considered likely.

Consequently, to correct this variability based on their physical features the scientific community has proposed to monitor the natural body light perception regulatory system (i.e., the eye). Reinhart [81] highlighted the human eye as the perfect light sensor and system actuator. As a personalized, naturally and highly evolved system, monitoring it could further calibrate visual comfort assessments (analogue to the body temperature in case of thermal comfort). For instance, the following approaches could be used:

- analysing the way the pupil size varies with respect to a certain lighting provision [82–85];
- monitoring pupil response when exposed to different lighting and work task type [86];
- screening the degree of eye opening [79];
- monitoring the view direction distribution and history under certain lighting provision [87];
- combining blinking, gaze direction, and pupil size variations records under a certain lighting environment [88]; and,
- analysing the frequency and extent of the facial muscles movement when exposed to expected discomfort glare [89].

The above-stated facts motivated a new approach towards visual comfort level evaluation, by coupling the building occupants' involuntary (i.e., body) response to the building management systems (BMS). The actual paradigm in which BMS systems are operated is contrasted on the basis of the averaged surveyed indoor physical parameters (see Figure 3). Therefore, hoping to reduce the gap between modelled and operating buildings' performance, by linking the occupant visual response to the BMS and using the data for more comprehensive understanding of the actual visual comfort. Consequently, increasing the satisfaction of occupants with the indoor luminous environment while also improving building energy efficiency, potentially reducing the risk of Sick Building Syndrome occurrence [90] and enhancing the livability of indoor spaces.

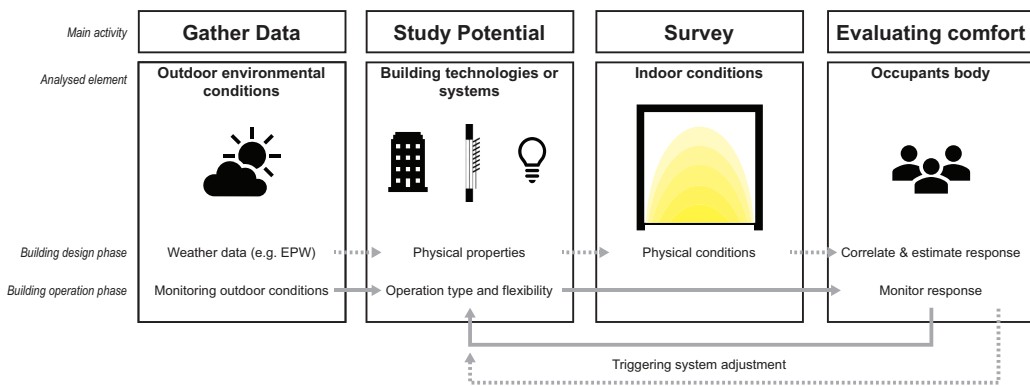

**Figure 3.** Description of the new concept for building components operation, shifting from monitoring the indoor physical conditions into monitoring the hosted occupant's body's physiological condition.

### 5.1. Eye Response to Light Variations

From the gathered literature, only a few metrics were found to directly assess the human eye response to the lighting environment, or indirectly account for the possibility of eye adaptation. These metrics are mostly based on the eye kinematics and only slightly on the quantity of light (either *E* or *L*). The identified metrics are summarized below:

- **Pupil diameter (*d*):** studied and condensed satisfactorily by De Groot and Gebhard [82]. An index is calculated based on the relationship between the pupil size and the luminance emitted towards it. It is grounded on a compendium of studies performed on human subjects for the medical research field of optometry. Equation (5) presents the most simplified form. Meanwhile, Equation (6) presented a greater correlation accuracy, especially when the corrected form of retinal illuminance ($E_R$) is used (Equation (7)) instead of the non-adjusted one (Equation (8)).

$$d = 4.9 - 3 \times \tan h[0.4(\log L + 0.5)] \tag{5}$$

$$\log E_R = \log L + 1.8614 - 0.000986 \times (\log L + 6.5) \tag{6}$$

$$\log E_R = 10 \times r^2 \times L \tag{7}$$

$$\log E_R = 10 \times r^2 \times L \times \left(1 - 0.0425 \times r^2 + 0.00067 \times r^4\right), \tag{8}$$

where:

$E_R$: retinal illuminance [trolands]
*d*: pupil diameter [mm]
*r*: pupil radius [mm]
*L*: luminance of the visual field [miliamberts]

- **Normalized pupil size (*NPS*):** Choi and Zhu [85] presented an analysis on how using a normalized pupil size value correlated to qualitative visual sensation communi-

cated by the tested human subjects. The pupil size was obtained using a dedicated pupilometer, and the *NPS* was computed using Equation (7).

$$NPS = \frac{Ps_i - Ps_{neutral}}{Ps_{neutral}} \times 100, \tag{9}$$

where:

$Ps_i$: pupil size at the current state [mm]
$Ps_{neutral}$: pupil size at a neutral lighting state [mm]

- **Task-evoked Pupillary Response (*TEPR*):** Klinger et al. [86] studied the pupil diameter variations when different task and lighting conditions were imposed to the human subjects, measured by head-mounted eye trackers and remote eye trackers using image processing (i.e., counting the pixels composing the pupil).
- **Pupil diameter-unified formula (*$D_U$*):** Watson and Yellott [91] presented a review on the previous research carried out to correlate the pupil response to lighting stimulus, highlighting the equation's accuracy, and their potential adjustments. Moreover, it proposed to combine different effects that tend to deviate the pupils' diameter calculation (i.e., *L*, age, field size and the monocular effect) to construct Equation (10) (applicable for occupants aged between 20–83).

$$D_U = D_{SD} + A \tag{10}$$

$$D_{SD} = 7.75 - 5.75 \left( \frac{(La/846)^{0.41}}{(La/846)^{0.41} + 2} \right)$$

$$A = (y - y_o) \times S$$

$$S = 0.021323 - 0.0095623 \times D_{SD},$$

where:

*L*: luminance [cd/m$^2$]
*a*: area [deg$^2$]
*y*: age of the human subject
$y_o$: minimum/reference/benchmark age

- **Degree of Eye Opening (*DEO*):** Yamin Garreton et al. [79] presented an analysis on how to use a normalized eye (or eyelid) height value. The pupil size was obtained using an analysis of images recording the state of the eyelid in correspondence with the maximum eye height (Equation (11)).

$$DEO = \frac{h_i}{h_{max}} \tag{11}$$

where:

$h_i$: eye/eyelid height [mm] or [pixels]
$h_{max}$: maximum eye/eyelid height [mm] or [pixels]

- **Gaze driven illuminance (*$E_{vg}$*):** Sarey Khanie et al. [92] exposed an analysis of how much would it change to compute the $E_v$ considering accurately occupant's gaze direction, aided by the use of Radiance images and evalglare tool [63,66].

Referring to the occupants' eye condition instead of the physical light quantity in the room would implicitly include the perceived lighting within the field of view and all the personal factors affecting the occupant perception. Using a correlation amongst the two would further bridge the gap between designed and monitored building performance. However, no guidelines have embraced any of the metrics described above or, incorporated them into an obligatory assessment method of the lighting design in buildings as an initial attempt to promote such an approach. A summary of the advantages and limitations of the above presented metrics have been condensed into Table 4.

**Table 4.** Critical analysis summary of visual perception metrics based on eye conditions, highlighting their benefits and drawbacks.

| Metric | Advantages | Disadvantages |
|---|---|---|
| $d$ | Accounts for directionality and human adaptation, highly responsive and reliable, light at eye known. | There is no established correlation with the occurrence of glare, no human variability, only one location, hard to measure and model. |
| $NPS$ | Accounts for directionality and human adaptation, refers to the saturation level of adaptability. | There is no established correlation with the occurrence of glare nor illuminance, only one location, hard to measure (especially for neutral condition) and model. |
| $TEPR$ | Measures pupil kinematics with high precision. | Highly intrusive measurement, no correlation with lighting environment performed, only pupil response recorded. |
| $D_U$ | Accounts for directionality, human adaptation and variability. Highly responsive and reliable. Light at eye known. | There is no established correlation with the occurrence of glare, only referred to one location and occupant, not applicable for young occupants, hard to monitor and model. |
| $DEO$ | Accounts for directionality and human adaptation, refers to saturation level of adaptability (not only pupil), correlation with $DGP$. | Scaling issues, only actions for eye-lid kinematics considered, no human variability considered, intrusive measurement, one location, hard to model. |
| $E_{vg}$ | Accounts for directionality and human adaptation, indirectly refers to adaptability (not only pupil), correlation with $DGP$. Light at eye known. | No human variability considered, intrusive measurement, one location, hard to model (requires known gaze behaviour). |

## 6. Emerging Occupant-Centred Visual Comfort Methods for Operating Building Lighting-Related Components

Boestra et al. [13] analysed the Health Optimisation Protocol for Energy-efficient Buildings database (HOPE, https://hope.epfl.ch/partners/partners-intro.htm (accessed on 23 December 2019)) that gathers post-occupancy comfort surveys on 60 office buildings with more than 6000 surveyed participants. Concluding that occupants frequently reported dissatisfaction with the lack or limited options for personal system control to adjust their surroundings to their demands and needs. The communication of occupants' intentions towards the BMS shall be personal and, if possible, highly responsive to the singular occupant or a group of occupants with similar needs.

Ensuring that the occupants' intended interaction correlates to the indoor environment's unpleasant conditions is vital for the building to acknowledge both the occupants' intentions and the indoor environment (physical) state. The use of sensors and automatized actuators has the potential to increase building operation performance. The availability of granular and real-time data enhances the response rate of any system towards more proficient results.

The industry has already demonstrated these informed control benefits, emphasizing the utility of sensors for acquiring critical variables' trends, process monitoring and control, but also sharing highlights on the advantages and weaknesses of current data-processing methodologies. Kadlec et al. [93] executed a review about available approaches to develop soft-sensors and to acquire and process data to build and train prediction models. Furthermore, they presented applications (e.g., on-line predictions, fault detection) and strategies on how to deal with common issues that arise when implementing them (e.g., missing data, outliers, diverse sampling rates and measurement delays). In addition, it is not only the implementation and monitoring of specific data that drives performance boost, it is also enhanced by: the collaborative interaction among systems or machines, the possibility of acknowledging and interpreting new experienced conditions for adaptation (executing "informed" decisions) [94]. However, it is challenging to achieve a smooth functioning under a collaborative data acquisition and processing; it requires first, that sensors' noise is identified and filtered accurately [95], and then effective algorithms that are able to prioritize and organize the effects of integrated actuators properly.

For example, Poli et al. [96] proposed and tested the use of sensors immersed within glazing components for monitoring indoor conditions and gathering occupants' response

to tune automatic blind control to regulate light influx. Rinaldi et al. [97] further analysed such components, testing a predictive model to later anticipate the occupants' intentions to regulate the window-blind states. Likewise, Karjalainen [98] presented a general conceptual work-flow to deal concurrently with heating, cooling, ventilation, lighting and automated blinds operation; by allowing occupants to communicate their interaction desires and by collecting data inputs of the weather forecast.

In more detail, recent research has concentrated on studying comfort from the singular occupant perspective. Occupant-based metrics have been proposed, as O'Brien et al. [99] did by normalizing results with occupancy density. This approach shift provides the advantage of understanding the real impact and needs of occupants' micro-conditions by having sensors and actuators close to the building occupants' workstations (in case of office buildings) and remotely controlled actuators from mobile or website applications (in case of residential buildings). From the studied actuators, their operation was found to be more in line with occupants' preferences (i.e., accurate response), daylighting proxies (parameters) were monitored at the occupants' work plane or from the occupant's view perspective.

### 6.1. Monitoring Occupants' Work Plane

When monitoring the occupants' work plane, the collected strategies were found to be mainly focused on surveying $E_h$ intensities and contrast ($U_o$). These strategies are easily applied on simulation-based design, likewise replicated in real-case scenarios. For instance:

- Jia et al. [100] presented a platform-based design framework for BMS considering data acquired from a camera, illuminance, temperature, $CO_2$, relative humidity (RH) and passive infra-red (PIR) sensors, and occupants' requests via mobile applications; managing decisions throughout the use of fuzzy logic to run HVAC and lighting appliances.
- Konstantakopoulos et al. [101] proposed the automation of shared lighting appliances through game theory for processing the building occupants' vote on lighting preferences.
- Reinhart [102] presented an approach of a light-switching and blind-operation model within a two occupants office, in which using simulation methods approach resulted in a 20% lighting energy reduction achieved by monitoring at every desk: occupants presence, light and radiation intensity, light appliances condition (ON/OFF) and predicting the possibility of turning them ON upon arrival [103], or during the day [22], and of turning them OFF [104].
- Gunay et al. [105] tested and studied the work-plane illuminance set-points of a control strategy for light-switching and blind operation on single office space through simulations with EnergyPlus with dynamically updated operation thresholds (modified by the probability models of user-interaction [22,103,104]). As a result, energy use reductions between 25–35% were achieved.
- Cheng et al. [106] studied the personal occupant visual comfort in parallel with energy savings (weighting cooling vs. lighting energy needs) and slat inclination angles of venetian blinds, monitoring lighting conditions with illuminance sensors on the desk and occupants interactions within a recreated office space facing east. The acquired datas was used to train and adjust the thresholds through a Q-learning process, leading to an energy saving potential up to 10%.
- Van De Meugheuvel et al. [107] proposed a different approach by studying ceiling-mounted sensor configuration through simulations in DIALux [108]. Using multiple intelligent dimmable luminaries equipped with occupancy and light sensors. These were calibrated at night to reach $E_h$ = 500 lux, to maintain desired lighting conditions working as stand-alone systems or as an integrated system network (for optimized luminaries activation also see Rubinstein et al., Caicedo and Pandharipande [109,110]).
- Jin et al. [111] studied the Indoor Environmental Quality (IEQ) by continuously monitoring an office area by a moving punctual reading (mounting sensors on a robot, including $E_h$). Doing so, it is possible to interpolate and map the total area conditions

with decent accuracy, given the surveyed values obtained while moving the robot throughout a predefined path.

Nevertheless, comparing the previously mentioned methods and more frequently utilized metrics, most of them analysed the light intensity on the work-plane, neglecting the light intensity and contrast perceived at eye level. Doing this is rather challenging; nonetheless, other methodologies have been elaborated based on the field of view to assess visual perception with higher accuracy, giving more importance to the light intensity falling on the human eye, its distribution around the space, and the personal occupant preferences (see Figure 4).

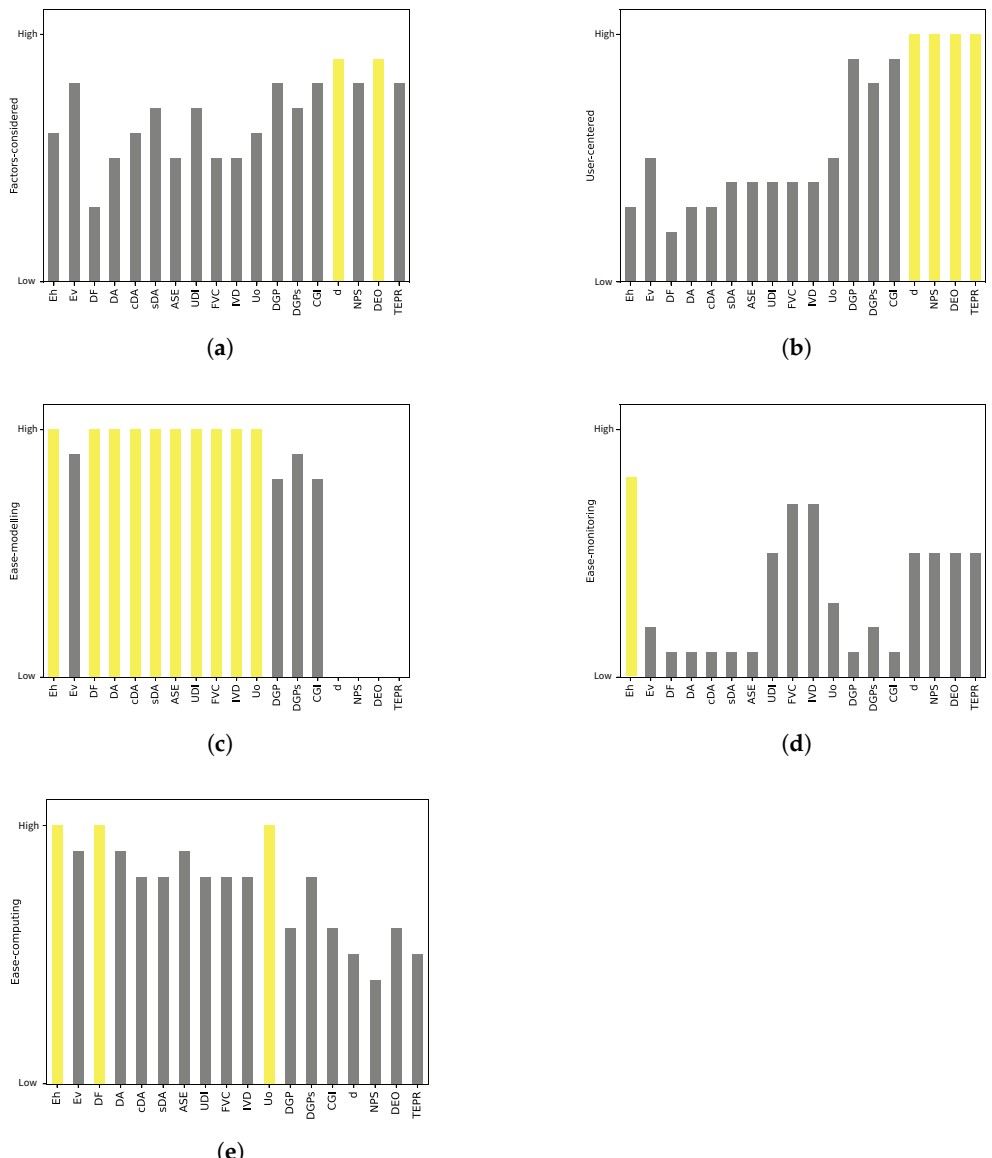

**Figure 4.** Direct comparison of the most utilized metrics under 5 different criteria, providing a subjective and experience-based rating, and highlighting in yellow the highest ranked in: (**a**) completeness of relevant factors considered (**b**) degree of occupant-centred approach (**c**) ease of modelling in the design phase (**d**) ease of monitoring in the operation phase (**e**) ease of computing.

## 6.2. Monitoring Occupants' Field of View

Recalling the information gathered in Tables 2–4 and the comparison made in Figure 4, personalized and, if possible, image-based analysis seems to be the most comprehensive and convenient way of assessing visual comfort. In this way, task lighting sufficiency,

contrast and glare risk can be evaluated simultaneously. For this purpose, different methods have been proposed and hereby summarized; however, difficulties arise considering the limitations of replicating simulation-based analysis within real scenarios.

- Image-based—Although these methods demand higher computation time, they are the ones capable of capturing different aspects that, as discussed previously, can alter the perception of the surrounding luminous environment.

    - Guerry et al. [112] proposed an innovative image-based methodology on the evaluation of contrast able to take into consideration visual impairments in health. The proposed method screens the produced images, identifying disturbing surfaces or elements within the field of view, based on a set threshold of contrast of luminance between two juxtaposed surfaces.

    - Based on the work of Glenn et al. [113] for the automotive sector, aiming to reduce night accidents due to wrong street lighting disposition and intensity; Zatari et al. [114] studied the possibility of automatizing this methodology for estimating glare risk, $L$ and $E_v$ in real-time by employing Charge Coupled Device (CCD) cameras mounted on a vehicle.

    - Based on the results obtained by Wienold and Christoffersen [63], Konis [115] presented an actual office space survey on 14 participants, coupling HDR images analysis (captured with CCD cameras oriented to the most frequent occupants' field of view) with polling station results on visual comfort rating; in order to test the accuracy in terms of glare rating estimation, with horizontal and $E_v$ based computations.

    - Using CCD cameras as a data acquisition system, it is possible to make, calibrate, and compute $E_v$ and $L$ from HDR images. These can be later used to monitor the interior conditions of an environment, as done by Parsae et al. [116] and Inanici [117]. The latter proposed the calculation by using the RGB values of an image and the $D_{65}$ reference illuminant, which after the proper calibrations, and accounting for the camera settings (e.g., exposure, focal length), obtained errors below 8% when testing the methodology under different light sources. On the other hand, Moeck [118] proposed a geometrical-based procedure to understand the illuminance value coming from a certain surface using CCD cameras and HDR images, acknowledging not only the camera settings, but also the distance between the objects and the camera. Under controlled conditions, the error obtained was below 5%.

    - Goovaerts et al. [119] tested the computation of $DGP$ from HDR images created from low-resolution camera pictures while monitoring $E_h$, to establish a venetian and roller blind control algorithm aiming to avoid the occurrence of visual discomfort. The $DGP$ was initially calculated, and then the strategy was tested on those users which interacted with the shading systems to increase daylight influx indoors. However, the $E_h$ was found to be underestimating the light intensity adaptation, and the $DGP$ to be overestimating the glare risk rating.

    - Motamed et al. [120] used two HDR ceiling vision sensors for monitoring work-plane illuminance (ceiling mounted) and $DGP$ (oriented towards occupants' visual display terminal (VDT)). This was done to lay down an advanced control algorithm for external roller blinds, which could in real-time, through fuzzy logic use, review the task illuminance compliance and glare risk from DGP values. By using this control logic to operate the external blinds confronted with a reference Eh at work-plane control, although all 30 subjects reported only slight variations on visual perception between control logics, energy needs from electrical appliances were reduced up to 31% compared to operate the roller blinds from monitored $E_h$ at the work plane.

- Wearable—The use of wearable devices for indoor monitoring was initially proposed by researchers assessing thermal comfort, later embraced by the research stream in visual comfort. They have resulted in very useful input for the building management

systems as they communicate a more localized and personalized measurement of environmental conditions. However, from the literature analysis, the collected strategies hereby presented are only applicable for assessing buildings during operation.

- Similar to the methodology employed by Sarey Khanie et al. [92] to define $E_{vg}$, Schneider et al. [121] developed a prototype of a head-mounted camera, for studying human gaze-behaviour. Using a video-oculography device (VOG) and a camera motion device, synchronously measure binocular eye positions at up to 600 Hz, it is possible to acknowledge where and what the user is seeing or looking at. These could ease any gaze driven analysis for visual comfort, but the degree of intrusiveness is yet too high to consider it for continuous monitoring.

- Yamin Garreton et al. [79] proposed Equation (11) for computing glare risk rating as $DEO$. Using an eye-tracker, they computed this metric and tested it on 20 subjects within a recreated office space. Based on the eye-lid position, it displayed a decent correlation with $DGP$ and $E_v$, under a wide range of lighting conditions (both low intensity/diffuse and high intensity/clear sky conditions). However, it still represents a highly intrusive method for continuous monitoring of the indoor environment lighting conditions and occupant perception.

- Choi and Zhu [85] investigated the potential of tracking the human pupil size to estimate visual sensation in office workplace environments. Profiting from its physiological capacity, as it reacts instantly with the visual environment variations. The tests were carried out on 20 subjects within a recreated office space, under different artificial lighting conditions and constant RH (avoiding dry-eye irritation effect). Pupil behaviour was monitored with a mobile pupilometer and the light intensity at the work-plane ($E_h$) with an illuminance meter. Although the intrusiveness is lower than in previously exposed studies, it is yet too restrictive to be applied as monitored visual comfort metric in operating buildings (requires users to wear the pupilometer constantly), and no shading effect from the eyelids was considered.

Thus, the above-presented new methodologies and the metrics condensed in Table 4, widen the possibilities to account for real occupants' lighting related needs or preference variability and bridge the gap between modelled and actual building performance. Visually comfortable ranges could be fine-tuned based on the actual reaction of the building occupants coupled with satisfaction surveys, including demographic factors that have been proven to affect the occupants' perception of the luminous environment. Furthermore, operating buildings could incorporate occupant-need responsive systems through clever monitoring networks incorporating some kind of occupant response tracking system.

## 7. Discussion

As mentioned in Section 3, acknowledging and incorporating the effect of occupants on building performance can contribute to narrowing down the performance gap between planned and actual building performance. Specially considering that based on building occupants demographic and physical features, occupants could be more susceptible to discomfort and, therefore, more inclined to intervene in building performance. Nevertheless, including this aspect in the definition of comfortable visual conditions for building design or operation is rather challenging. This variability is still unconsidered in current building guidelines, standards and methodologies for the assessment of the indoor luminous environment. Contrary to thermal comfort assessment, only general considerations proposing higher illuminance intensities have been made for visually impaired occupants [50,122].

Standards have been passive on including complex assessments for visual comfort, and established detailed approaches are mainly related to those proposed by certification procedures (e.g., LEED, BREAM). Nonetheless, the challenge remains on the complexity of associating a spatial and time-based approach to the sensitivity of the single building occupant. Moreover, when a diverse occupancy is considered, in complex environments, or

in operating buildings, there are few possibilities to collect in parallel all their perceptions and needs to be able to satisfy all of them contemporarily.

The challenge presented above requires different but integrated solutions. One potential solution could be by starting to complement qualitative survey data with objective physiological response on the ratings given to different luminous environments. This would provide more accurate definitions of the comfortable visual environment; it will also enrich the knowledge and provide further certainties on the factor's affection visual perception. Then, such definitions can be diversified by testing similar conditions for representative subjects of each of the physiological factors that can affect their perception. Having these information enables the fine-tuning of existing metrics, or the proposal of new ones, to accurately estimate visual comfort in a single point for different space uses and demographic contexts. Thus, different simulation workflows can be optimized to maintain admissible computation time to perform time-based and spatial visual comfort assessments utilizing such metrics. Finally, non-intrusive monitoring strategies can be designed to run in operating buildings, targeting directly or indirectly the parameters defined to compute the developed or adjusted metrics.

The use of the body response as a proxy (i.e., eye response) to monitor visual stimulus can be one holistic solution. In fact, research has already been done on its relationship with environmental parameters ($E_R$, $E_p$ and $L$) and in some of the factors that could impact the perception of the luminous environment (excluding those related to the emotional dimension). Using the body response is personalized for each occupant and it can be monitored on an operational building (taking care of the level of intrusion and privacy).

*Limitations*

Numerous visual comfort assessment methods and strategies have been collected, reviewed, and compared, finding as a potential solution the need for a more diversified and personalized metric, or number of metrics, and a more occupant-centred assessment. However, the collection of the literature was conducted by targeting diffused and innovative methods of visual comfort assessment (on an expert criteria basis). Additionally, literature that contained visual comfort analysis was put aside if a higher focus was given to thermal comfort or overall building energy use analysis. Thus, the literature collection procedure could have resulted in being unintentionally biased. Future updates to this work could be hampered as a generic query was utilized and variable filters were imposed. Nevertheless, the latest findings support that the course that visual comfort research is taking is aligned with what has been presented.

## 8. Conclusions and Further Developments

The present analysis concentrated on: (i) scrutinizing and comparing visual comfort assessment works and established procedures, to deduct advanced methods to characterize, monitor and rate the luminous environment. The most used metrics to rate visual comfort were compared and their strengths and weaknesses were presented, based on the identified need to capture a more accurate visual perception; (ii) Different methods and approaches to monitor the luminous environment and estimate the visual comfort were scrutinized for building design and operation. Their shortcomings were identified and potential improvements were proposed, in order to incorporate important missing influential factors such as the ones gathered by Persont et al. [36]. Consequently, tracking the eye response and correlating this response with illuminance was identified as a promising approach to enhance the accuracy to estimate visual comfort, to improve lighting design in buildings and thus, to cover the gap between expected and actual building performance.

However, further research is necessary following what has been done with the metrics presented in Section 5.1. In this regard, the actual eye response to light variations of both healthy and visually impaired occupants should be studied to attain more accurate visual comfort ratings. Obtained data could then be used to adjust the thresholds of the existing metrics (or, if deemed necessary, proposing new ones) to account for such physiological visual perception differences. As a result, better limit thresholds for building

systems operation could be defined, while more appropriate occupancy definition and better monitoring of indoor lighting environments could be executed.

A proposal for a novel approach to evaluating a luminous indoor environment based on human-centric matrices could be defined. In general, it could be summarised as having a more detailed definition of the occupant inhabiting the building, better anticipating their preferences for luminous indoor conditions, and consequentially reducing unexpected building performance due to visual discomfort. In order to achieve the stated goals:

- In the instance of existing buildings for monitoring:
    - Transferring and integrating novel and growing technologies such as computer vision, image processing, scene understanding, and deep learning can further enable monitoring and operation strategies. They can help identify occupancy type (including occupant characteristics) that can be feed to the BMS for operational profiling. They can potentially decrease the level of intrusiveness for monitoring body response. Or, making components responsive, with data-training models able to capture occupant interactions and predict occupant needs [123].
    - Lean sensorial network and internet of things implementation have great potential to better interpret and communicate occupancy, overall indoor conditions and personalized task area conditions [124,125]. If these are fully integrated into a flexible plant system, personalized or localized building responses can be activated for providing, overall, better indoor environmental quality.
    - Alternatively, qualitative surveys and post occupancy evaluations can be utilized in real-time for requesting to the building specific operation adjustments. Comparable to the upgrade of a traditional manual control enhanced by digital technologies [126].
- In the instance of both existing and new buildings for design:
    - Data acquired in occupancy studies could serve better for understanding building occupant preferences in tendency in occupation, lighting appliances and blinds activation schedules, and space occupation (complementing missing information according to the specific LOD) (i.e., occupant modelling) [10]. Then, these data can be used for a more accurate simulation output with realistic occupant behaviour [127].
    - Diversified post-occupancy surveys on perception of the luminous environment (or visual comfort rating) could be used for training algorithms and defining prediction models resulting in average virtual occupant typology. Or, with such databases, occupants' preferences profiling can be carried out to adjust traditional or new visual comfort performance metrics [128].
    - Furthermore, utilizing personalized and complex visual comfort performance metrics would yield the analysis more complete. These could include demographics and spatial dependent factors by fine-tuning the correlating between body response (in this case the eye for visual comfort, using $d$ and/or $DEO$) with environmental parameters (i.e., $E_v$, $E_R$ or $E_p$, and $DGP$).
    - To assure that such calculations are done within applicable computational time, annual and spatial simulations could be: (i) performed using daylight coefficients, or cubic illuminance, instead of rendered-image-based analysis [72]; (ii) they can be structured in a way that these can be parametric so they can be run in parallel through cloud computing services [74]; (iii) Or, as a mid-term solution, simulations can be decomposed in such a way that many smaller and more specialized cores can be used to process the computational task (delivering higher computational performance) in GPUs [75].

In conclusion, this work is foreseen as a catalyzer to modernize the traditional way of assessing visual comfort, estimating visual perception and rating the lighting provision. Transiting from monitoring/studying the environment onto monitoring/studying the occupant response or occupant interactions itself. Motivating the inclusion of the occupant

demographic features could render the analysis completer and more accurate. Thus, boosting the capacity of designer to predict the actual performance of buildings and to reduce the performance gap issues stemming from the occupants visual dissatisfaction.

**Author Contributions:** Conceptualization, J.D.B.C. and T.P.; methodology, J.D.B.C. and T.P.; valida­tion, J.D.B.C., T.P., M.K. and G.L.; formal analysis, J.D.B.C.; investigation, J.D.B.C., T.P., M.K., G.L., A.G.M. and A.S.; resources, J.D.B.C., T.P., A.G.M. and A.S.; data curation, J.D.B.C., T.P., M.K., G.L., A.G.M. and A.S.; writing—original draft preparation, J.D.B.C.; writing—review and editing, J.D.B.C., T.P., M.K., G.L., A.G.M. and A.S.; visualization, J.D.B.C. and T.P.; supervision, T.P., M.K., G.L. and A.G.M.; funding acquisition, J.D.B.C., T.P., M.K., G.L., A.G.M. and A.S. All authors have read and agreed to the published version of the manuscript.

**Funding:** This research was funded by ABC Department—Politecnico di Milano (finanziamento delle attività per la valorizzazione della ricerca e TM DABC). The author M. K. acknowledges the financial support from the Slovenian Research Agency (research core funding No. P2-0158). The author G.L. acknowledges the financial support from the Norwegian Research Council (research project FRIPRO-FRINATEK no. 324243 HELIOS).

**Institutional Review Board Statement:** Not applicable.

**Informed Consent Statement:** Not applicable.

**Data Availability Statement:** Not applicable.

**Conflicts of Interest:** The authors declare no conflict of interest.

## Abbreviations

The following abbreviations are used in this manuscript:

| | |
|---|---|
| $ASE$ | Annual Sun Exposure |
| BEM | building energy modelling |
| $BGI$ | British Glare Index |
| BMS | Building Management System |
| CCD | Charge Coupled Device |
| $cDA$ | continuous Daylight Autonomy |
| $CGI$ | International Commission on Illumination Glare Index |
| $D_U$ | Pupil diameter unified formula |
| $DA$ | Daylight Autonomy |
| $DEO$ | Degree of Eye Opening |
| $DF$ | Daylight Factor |
| $DGP$ | Daylight Glare Probability |
| $DGP_s$ | simplified Daylight Glare Probability |
| $E$ | Illuminance |
| $E_h$ | Horizontal illuminance |
| $E_p$ | Pupil illuminance |
| $E_R$ | Retinal illuminance |
| $E_v$ | Vertical illuminance |
| $E_{vg}$ | Gaze driven illuminance |
| EN | European Committee for Standardization (CEN) standard |
| $FVC$ | Frequency of Visual Comfort |
| HDR | High Dynamic Range |
| HVAC | Heating, Ventilation and Air Conditioning |
| IEQ | Indoor Air Quality |

| | |
|---|---|
| *IVD* | Intensity of Visual Discomfort |
| KPI | Key Performance Indicator |
| *L* | Luminance |
| *NPS* | Normalized pupil size |
| RGB | Red, Blue and Green colour space |
| RH | Relative humidity |
| *sDA* | spatial DA |
| *TEPR* | Task-evoked Pupillary Response |
| *U_o* | Daylight Uniformity |
| *UDI* | Useful Daylight Illuminance |
| VDT | Video Display Terminal |

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
