# Peer review of "Current Trajectories and New Challenges for Visual Comfort Assessment in Building Design and Operation: A Critical Review"

_applsci, doi:10.3390/app12063018_

Round 1

Reviewer 1 Report

This work focuses on reviewing literature findings on how the common design approach deviates from real building performance, particularly failing to prevent visual disturbances that can trigger inefficient operation of building systems. Additionally, it is highlighted that redesigned visual comfort assessment methods and metrics are required to bridge the gap between the lighting environment ratings computed and surveyed. One possibility is to consider such physiological features that persuade lighting experience. Finally, it was deducted that it is important to target the occupants’ eye response to calibrate limit thresholds, propose occupant profiling and that it is convenient to monitor continuously the occupants’ perception of indoor lighting conditions.

This paper discusses an excellent aspect that directly contributes to the concept of sustainability. However, I think the authors should justify the rationale of this research by highlighting the concept of "livability". 

Reducing energy consumption is indeed the main aspect of sustainability, however, this shall be done while increasing the people's quality of life (livability). This paper is indeed contributing to the concept of livability which eventually increases sustainability. Therefore, please build your argument on this base. 

Author Response

Dear Reviewer,

On behalf of the authors, I would like to thank you for the time and effort dedicated to make the review of our manuscript. And also, on the positive comments that were made on the quality of the work and the resulting outcome.

We are certain that with your contribution and comments, the quality of the work submitted is guaranteed.

The English of the manuscript has been revised (verifying that grammar, spelling and typing errors were resolved) to make the text more organic, fluent, and more efficient to communicate the expressed ideas.

Additional references have been added, to enrich and further support the assertions made in the document.

In specific, your specific comments were resolved as follows:

  • This paper discusses an excellent aspect that directly contributes to the concept of sustainability. However, I think the authors should justify the rationale of this research by highlighting the concept of "livability". Reducing energy consumption is indeed the main aspect of sustainability, however, this shall be done while increasing the people's quality of life (livability). This paper is indeed contributing to the concept of livability which eventually increases sustainability. Therefore, please build your argument on this base. 
    • The concept of livability is indeed fundamental for the design of indoor environments. This was assessed by enhancing the satisfaction of building occupants with the indoor environment. In addition, the concept of livability, quality of life and of the space for occupants was introduced while highlighting the reduction of potential occurrence of the sick building syndrome.

Reviewer 2 Report

Review Report: applsci-1639813

Journal: Applied Sciences (ISSN 2076-3417)

Manuscript ID: applsci-1639813

Type: Article

Title: Current trajectories and new challenges for visual comfort assessment in building design and operation. A critical review

Authors: Juan Diego Blanco Cadena * , Tiziana Poli , Mitja Kosir , Gabriele Lobaccaro , Andrea Giovanni Mainini , Alberto Speroni

Section: Energy Science and Technology

Special Issue: Beyond Energy Efficiency in Architecture. New Challenges and Research Trajectories for Buildings and the Built Environment

------------------------------------------------

Dear Colleagues,

first of all, I would like to congratulate the authors for their work in which we find a review around visual comfort in buildings. Since many years, we can see two main philosophies: saving energy (and reduce Carbon Impact) and/or the human centric point of view which are dissociate but not incompatible. Your review focuses mainly on the last one which is certainly the first to be considered. In this paper, you consider the whole aspect from the metrics to rate visual comfort to the way for monitoring the luminous environment. This paper proposes as a conclusion a new human centric approach to evaluate it.

it is clear that there is not a single ideal lighting solution in an environment because the lighting depends on the type of activity of the occupant (rest, leisure, work) as much as on his own visual capacities dependent on especially his age. Indeed, the influence of the color temperature of lighting on the production of melatonin by the pineal gland and therefore on attention and alertness, for example, is well known. A high color temperature will be more suitable for stimulating intense intellectual activity, while low color temperatures will be suitable for resting and falling asleep. Considering only these two factors (activity and age), only adaptive lighting can meet the needs of the occupant.

In general, the whole article is well structured, well written and of obvious interest. There are no substantive remarks that could pose a problem for its publication.

Nevertheless, as a reviewer, I will try to improve and complete your work by providing additional information and correcting some minor errors.

Indeed, the article can be supplemented by a few references and it contains many typing errors.

I am going to do a linear reading of your article by mentioning more or less important remarks but which will have to be taken into account when making corrections.

Page 1:

Typo in the name “Gabriele Lobacarro”

L22 : 2 times “to also include”

Paragraph L34/L39 (but also line 490/492 or 523):

The following reference can be useful to illustrate this paragraph. Indeed, this paper presents a low cost (useful in many poor countries where both the need to save energy and the price of appliance control management are important factors) automation system to control light and blinds in warm countries where well-being is more linked to ambient temperature and where the visual comfort is a second parameter.

But this can be also useful and cited in page 17 in the paragraph 6.1 with Cheng et al (line 523) or previously in line 490 where you talk about blind control or in line 492 where cooling and lighting are mentioned.

  1. Chekired et al., "Low Cost Automation System for Smart Houses based on PIC Microcontrollers," 2020 IEEE International Conference on Environment and Electrical Engineering and 2020 IEEE Industrial and Commercial Power Systems Europe (EEEIC / I&CPS Europe), 2020, pp. 1-5, doi: 10.1109/EEEIC/ICPSEurope49358.2020.9160808.

As you see, this reference can be mentioned in many part of your paper and will find fully its place in your topic.

Page 6

Line 143 : 2 times “the” occupancy

Paragraph 3.3

In this paragraph, you can cite the reference below which deals with visual perception in the elderly. E. Guerry et al. provide an innovative methodology on the evaluation of contrasts taking into account visual impairments in health establishments adapted to the elderly and this is perfectly related to the subject of this paragraph.

  1. Guerry, C.D. Galatanu, L. Canale and G. Zissis, "Luminance Contrast Assessment for Elderly Visual Comfort Using Imaging Measurements", 12TH International Conference Interdisciplinarity in Engineering (INTER - ENG 2018) Procedia Manufacturing, vol. 32, pp. 474-479, 2019.

L178 :

In paragraph 4, as you mention in line 198, contrast is an important point of visual comfort. The absence of contrast does not allow the eye to focus easily and induces visual fatigue. This point could be a little more emphasized in the introduction to this paragraph.

Line 185, missing space between the coma and “it”.

Line 230: the sentence beginning with the subject "the average of the building" should be rewritten. You probably mean "average building DF"

Line 277: “Radiance” : no need an uppercase

Line 342: “Ladybug tools”. This platform is not universally known. You should add some information’s or something to introduce this before.

Line 401: typo / double “a” new approach…

Line 403: sentence “Contrasting the actual paradigm in which systems are operated from the average surveyed indoor physical parameters (see Figure 3).” This sentence must be rewritten. This is only the start of a sentence without verb.

Line 405: to the BMS “and” : typo/space inside de word

Line 409: 2 times “occurrence”

Line 458: missing “n” / an obligatory

After Line 543: something wrong in Subtitle of Figure 4 (character size or space between lines)

Line 579: avoid repetitions if possible (3 times the same word “mounted”)

Line 587: reference “F. Chekired et al.” can be mentioned also here.

Line 668 and 669 : same typo problem on the word “luminous” (2 times)

Line 670 : typo : “identified”

Line 930: “tools, L. Pollination” ; is it possible to add information (web link…? Freeware? Editor? Is it open source? Etc…)

Line 1043: respect the same typo : All name with first character in uppercase and others in lowercase

----------------------------------------------------

Conclusion:

The article is clear and well written. It can be accepted for publication with recommendations to take into consideration the few remarks above.

It can also serve as a reference and be used by architects during building construction as well as for interior design. I renew my congratulations to the authors for their work.

Best Regards.

Author Response

Dear Reviewer,

On behalf of the authors, I would like to thank you for the time and effort dedicated to make the review of our manuscript. And also, on the positive comments that were made on the quality of the work and the resulting outcome.

We are certain that with your contribution and comments, the quality of the work submitted is guaranteed.

The English of the manuscript has been revised (verifying that grammar, spelling and typing errors were resolved) to make the text more organic, fluent, and more efficient to communicate the expressed ideas.

Additional references have been added, to enrich and further support the assertions made in the document.

In specific, your specific comments were resolved as follows:

  • improve and complete your work by providing additional information and correcting some minor errors.
    • Additional information has been included in the article to enrich the content, and the discussion quality on the challenges ahead of visual comfort assessment. For instance, ASE was moved form Table 2 to Table 3 given that in the ttext it was catalogued as a discomfort measure; in Table 2, “n/a” was included to state that there were no guidelines found to be associated to such metrics; in table 4 DU and Evg were inserted to include their advantages and disadvantages as well. Also, some references were added where useful to support an idea (i.e. for sick building syndrome occurrence in line 418 Al horr, et al. (2016) “Impact of indoor environmental quality on occupant well-being and comfort: A review of the literature” has been included)
    • Minor errors have been amended (typos, spelling and grammar).
  • the article can be supplemented by a few references, and it contains many typing errors.
    • Additional information has been included in the article to enrich the content, and the discussion quality on the challenges ahead of visual comfort assessment. For instance, ASE was moved form Table 2 to Table 3 given that in the ttext it was catalogued as a discomfort measure; in Table 2, “n/a” was included to state that there were no guidelines found to be associated to such metrics; in table 4 DU and Evg were inserted to include their advantages and disadvantages as well. Also, some references were added where useful to support an idea (i.e. for sick building syndrome occurrence in line 418 Al horr, et al. (2016) “Impact of indoor environmental quality on occupant well-being and comfort: A review of the literature” has been included)
    • Minor errors have been amended (typos, spelling and grammar).
  • Typo in the name “Gabriele Lobacarro”
    • The name of the author has been corrected as “Gabriele Lobaccaro”
  • Line 143 : 2 times “the” occupancy
    • Repeated word has been removed
  • Paragraph 3.3 - In this paragraph, you can cite the reference below which deals with visual perception in the elderly. E. Guerry et al. provide an innovative methodology on the evaluation of contrasts taking into account visual impairments in health establishments adapted to the elderly and this is perfectly related to the subject of this paragraph. Guerry, C.D. Galatanu, L. Canale and G. Zissis, "Luminance Contrast Assessment for Elderly Visual Comfort Using Imaging Measurements", 12TH International Conference Interdisciplinarity in Engineering (INTER - ENG 2018) Procedia Manufacturing, vol. 32, pp. 474-479, 2019.
    • It was added instead to section 6.2, as it can be coupled with field of view analysis to establish the degree of contrast that occupants perceive. This was added as follows: “Guerry et. al proposed an innovative image-based methodology on the evaluation of contrast able to take into consideration visual impairments in health. The proposed method screens the produced images, identifying disturbing surfaces or elements within the field of view, based on a set threshold of contrast of luminance between two juxtaposed surfaces.”
  • L178 - In paragraph 4, as you mention in line 198, contrast is an important point of visual comfort. The absence of contrast does not allow the eye to focus easily and induces visual fatigue. This point could be a little more emphasized in the introduction to this paragraph.
    • This contribution was intended as the quality of the light. And, the concept of induced fatigue was included while describing the importance of both intensity and quality in Section 4.1
  • Line 185 - missing space between the coma and “it”.
    • The missing space has been added.
  • Line 230 - the sentence beginning with the subject "the average of the building" should be rewritten. You probably mean "average building DF"
    • The sentence has been modified as: “the average DF across the analyzed surfaces of the building is typically requested to be greater or equal to 2%”
  • Line 277 - “Radiance” : no need an uppercase
    • The uppercase has been converted into lowercase
  •   Line 342 - “Ladybug tools”. This platform is not universally known. You should add some information’s or something to introduce this before.
    • A citation has been done for ladybug tools and Pollination. Differentiating between the cloud computing platform (Pollination) and the available plug-ins (Ladybug Tools).
  • Line 401 - typo / double “a” new approach…
    • The duplicated word has been removed
  • Line 403 - sentence “Contrasting the actual paradigm in which systems are operated from the average surveyed indoor physical parameters (see Figure 3).” This sentence must be rewritten. This is only the start of a sentence without verb.
    • The phrase has been changed to highlight that it is referred to the way buildings, or the building management systems are operated. It now reads as follows: “…Contrasting the actual paradigm under which BMS systems are operated on the basis of the averaged surveyed indoor physical parameters…”
  • Line 405 - to the BMS “and” : typo/space inside de word
    • The additional space was removed
  • Line 409 - 2 times “occurrence”
    • The repeated word has been removed
  • Line 458 - missing “n” / an obligatory
    • The typo error has been resolved
  • After Line 543 - something wrong in Subtitle of Figure 4 (character size or space between lines)
    • The caption of Figure 4 has been corrected; it was a mistake on the latex template.
  • Line 579 - avoid repetitions if possible (3 times the same word “mounted”)
    • Only “ceiling mounted” has been used, while the other two were removed to avoid being redundant
  • Line 587 - reference “F. Chekired et al.” can be mentioned also here.
    • It was impossible to trace back to Chekired et al. to to be included in the section. However, one of the references previously used was integrated into the section.
  • Line 668 and 669 - same typo problem on the word “luminous” (2 times)
    • The text was revised and corrected to avoid the word repetition, making the text more fluent.
  • Line 670 - typo : “identified”
    • The typo has been corrected.
  •   Line 930 - “tools, L. Pollination” ; is it possible to add information (web link…? Freeware? Editor? Is it open source? Etc…)
    • The whole name of the cloud simulation platform and the url was included
  • Line 1043 - respect the same typo : All name with first character in uppercase and others in lowercase
    • The references database was corrected and updated to resolve this issue.